# Mechanistic Insights on Microbiota-Mediated Development and Progression of Esophageal Cancer

**DOI:** 10.3390/cancers16193305

**Published:** 2024-09-27

**Authors:** Kyaw Thu Moe, Kevin Shyong-Wei Tan

**Affiliations:** 1Biomedical Sciences, Newcastle University Medicine Malaysia, Iskandar Puteri 79200, Johor, Malaysia; 2Laboratory of Molecular and Cellular Parasitology, Health Longevity Translational Research Programme, Department of Microbiology and Immunology, Yong Loo Lin School of Medicine, National University of Singapore, 5 Science Drive, Singapore 117545, Singapore

**Keywords:** esophageal cancer, oral microbiota, microbial dysbiosis, carcinogenic metabolites, chronic inflammation

## Abstract

**Simple Summary:**

Esophageal cancer (EC) is a serious global health problem with increasing incidence and mortality rates. Aside from smoking and alcohol consumption, recent research has linked oral microbiota with EC carcinogenicity. This review highlights the relationship between oncology and the associated pathogens, with a focus on the common species of harmful bacteria that promote cancer, such as *Porphyromonas gingivalis* and *Fusobacterium nucleatum*, and the potentially essential mechanisms underlying these phenomena, e.g., chronic inflammation, microbial dysbiosis, and carcinogenic substance production. Additionally, the review explores how long-term gastroesophageal reflux disease may alter microbiota structure and discusses the feasibility of bacteriotherapy to modulate microbiota–immune system interactions for EC prevention.

**Abstract:**

Esophageal cancer (EC) is one of the most common malignant tumors worldwide, and its two major types, esophageal adenocarcinoma (EAC) and esophageal squamous cell carcinoma (ESCC), present a severe global public health problem with an increasing incidence and mortality. Established risk factors include smoking, alcohol consumption, and dietary habits, but recent research has highlighted the substantial role of oral microbiota in EC pathogenesis. This review explores the intricate relationship between the microbiome and esophageal carcinogenesis, focusing on the following eight significant mechanisms: chronic inflammation, microbial dysbiosis, production of carcinogenic metabolites, direct interaction with epithelial cells, epigenetic modifications, interaction with gastroesophageal reflux disease (GERD), metabolic changes, and angiogenesis. Certain harmful bacteria, such as *Porphyromonas gingivalis* and *Fusobacterium nucleatum*, are specifically implicated in sustaining irritation and tumor progression through pathways including NF-κB and NLRP3 inflammasome. Additionally, the review explores how microbial byproducts, including short-chain fatty acids (SCFAs) and reactive oxygen species (ROS), contribute to DNA harm and disease advancement. Furthermore, the impact of reflux on microbiota composition and its role in esophageal carcinogenesis is evaluated. By combining epidemiological data with mechanistic understanding, this review underscores the potential to target the microbiota–immune system interplay for novel therapeutic and diagnostic strategies to prevent and treat esophageal cancer.

## 1. Introduction

Upper gastrointestinal cancers, including gastric and esophageal, have high incidence and mortality worldwide. According to the World Health Organization, cancer is now the top or second leading cause of death among individuals under 70 in over 90 countries, rising to the third or fourth position in an additional 22 nations [1]. While the root causes of several cancers like smoking for lung cancer or alcohol combined with occupational hazards for upper gastrointestinal cancers are well established, the etiology of EC is multifactorial.

EC represents the seventh most common malignancy diagnosed worldwide, with more than 470,000 new cases per year [2]. The two fundamental histological subtypes of EC are EAC and ESCC. EAC is predominantly found in developed countries and is strongly associated with GERD, Barrett’s esophagus (BE), obesity, low fruit and vegetable intake, and smoking [3]. On the contrary, ESCC is more common in developing countries, with risk factors including alcohol consumption, smoking, and poor oral health [4].

East Asia has one of the highest incidence and mortality rates for gastrointestinal cancers, with a great disease burden. In 2020, East Asia reported nearly half of the world’s newly diagnosed gastrointestinal cancer cases, with significant contributions from countries like Mongolia, Japan, China, South Korea, and North Korea [5]. The mortality rates reflect the high incidence, underscoring the urgent need for effective prevention, early detection, and treatment strategies. These divergences may be attributable to varying environmental, lifestyle, and dietary elements.

EC is generally diagnosed at a late stage, resulting in poor prognosis and low survival rates. The five-year survival rate of most countries is around 15 to 25 percent, thus there is an urgent need for new avenues of prevention, risk stratification, and early detection [6]. Emerging evidence further suggests the possible involvement of the upper digestive tract microbiota in the etiology of EC, especially in the case of EAC’s rising incidence in developed countries.

The highly diversified microbiota of the upper digestive tract includes mutualists, commensals, and the pathogens that actively promote carcinogenesis by activating toll-like receptors (TLRs) or resisting carcinogenesis as the organisms can synthesize vitamins or provide barriers to carcinogenesis. Cross-sectional studies have shown that there is a significant difference in the microbiota of people with GERD, BE, EAC, esophageal squamous dysplasia, and ESCC compared to the healthy controls [7]. Furthermore, diseases in the oral dysbiosis context such as periodontitis have been found to be related with a higher risk of EC [8].

Recent investigations have highlighted the crucial contribution of the human microbiota in cancer development. The human mouth hosts a diverse array of microorganisms including bacteria, archaea, fungi, and viruses. Numerous analyses have demonstrated a close connection between oral microbes and gastrointestinal tumors. These microorganisms may contribute to carcinogenesis through mechanisms such as the production of carcinogenic substances, chronic inflammation, and altered cell metabolisms [8].

### 1.1. Risk Factors

#### 1.1.1. Common Non-Modifiable Risk Factors

Ethnicity has a considerable influence on the risk of gastrointestinal cancers, and this varies between countries. For instance, in the USA, the East Asians and Hispanics as well as Blacks experience a higher risk of stomach cancer than the non-Hispanic white population [9]. East Asians and Black individuals are also at a higher risk of liver cancer relative to the white individuals with chronic hepatitis C virus infection and cirrhosis [10]. ESCC is predominant in East Asia, whereas EAC is common in the West [11]. Age is a substantial risk factor for gastrointestinal cancers, with almost all cases occurring in persons aged over 50 years [12]. In addition, men are more at risk of getting most gastrointestinal cancers than women, implying that the cancer’s etiology is likely more frequent in some groups or occurs earlier in life for the males [13]. Also, a family history of gastrointestinal cancers increases the risk, probably due to the inheritance of genetic features and sharing the environmental risk factors. Specifically, mutations in the tumor protein p53 (TP53), cyclin-dependent kinase inhibitor 2A (CDKN2A), and neurogenic locus notch homolog protein 1 (NOTCH1) increase the risk of EC [14].

#### 1.1.2. Common Modifiable Risk Factors

Smoking is a major modifiable lifestyle risk factor for the occurrence of gastrointestinal cancers in East Asia and account significantly for the colorectal cancer incidence and mortality burden among Chinese men [15]. Another major risk factor is drinking alcohol, which can lead to a rise in colorectal, stomach, esophageal, and gallbladder cancers [16,17,18]. The other risk factors are unhealthy dietary habits including the low intake of dairy and whole grains and the high consumption of salt along with processed foods. The positive associations with colorectal cancer are strong for red or processed meat, but diets rich in fruits, vegetables, vitamins, and fiber are generally protective [19]. Physical inactivity and sedentary behavior have been linked positively to various cancers, including colorectal, stomach, liver cell gall bladder, and pancreatic, while regular physical activity has been associated with a decreased risk of different cancers [20]. In Asia, obesity is defined as a BMI of ≥25 kg/m^2^ and has been associated with gastrointestinal cancers [15]. Metabolic disorders, such as type 2 diabetes and hypertension, are also associated with increased risks of these cancers in East Asia [21].

#### 1.1.3. GERD and BE

GERD results from the dysfunction of the lower esophageal sphincter (LES); as a result, acidic contents from the gastric region return to the esophagus. The continuous acid exposure to the esophageal lining irritates the mucosa and causes damages that result in inflammation. Subsequently, the continuous cycle of epithelial damage due to acid exposure and reflux initiates pro-inflammatory pathways that involve crucial cytokines such as IL-1β, IL-6, and tumor necrosis factor-alpha (TNF-α) [22]. These cytokines attract immune cells that further fuel inflammation and, sporadically, these cells damage the LES. This situation heightens acid reflux.

In GERD, the esophageal microbiota is characterized by dysbiosis, and Gram-negative bacteria such as *Prevotella* and *Fusobacterium* are the most abundant [23]. These bacteria produce lipopolysaccharides (LPSs), which triggers the nuclear factor kappa-light-chain-enhancer of activated B cells (NF-κB) pathway via TLR-4 in the epithelial cells [24]. The NF-κB subsequently triggers the release of pro-inflammatory cytokines (Figure 1). The concentration of these cytokines stays high, and the above sequence of events is repeated, leading to chronic inflammation and the incidence of further damage. Environmental factors, such as a high-fat diet, smoking, and alcohol consumption, intensify the continuation of these phenomena [25]. The chronic oxidative stress and ROS result in DNA damage and the initiation of another inflammatory episode.

BE develops due to consistent acid and bile reflux to the esophageal lining. In response to these environmental factors, the esophageal squamous epithelium undergoes a metaplastic transformation into a columnar epithelium, an adaptive measure to resist further acid exposure. However, the columnar epithelial cells are at a high risk of leading to EAC [26]. The continuous inflammation and microbial activities activate TLR-4, hence the presence of NF-κB in the columnar cells. As a result, chronic inflammatory cytokines such as IL-1β and IL-6 are released. The chronic oxidative stress from ROS and the refluxate damage the columnar DNA for the metaplastic alterations to develop. The risk of the occurrence is further escalated by genetic and epigenetic alterations such as p53 mutations and pro-inflammatory proteins like COX-2 expression [27]. This situation is aggravated by smoking and obesity, which continue to produce an inflammogenic microenvironment. Therefore, the pathogenesis of BE and potential development to EAC is based on GERD-induced chronic inflammation and microbiota-induced dysbiosis.

#### 1.1.4. Highlighted Risk Factors in East Asia

In East Asia, the population is at a higher risk of stomach cancer, with a higher prevalence of *Helicobacter pylori* infection compared to Western countries, as a significant cause of many gastric cancer cases [28]. A distinctive risk factor for biliary tract, liver, and gall duct cancer is the liver fluke *Clonorchis sinensis*, which is a significant risk in the Guangdong and Guangxi provinces in Southern China and the Heilongjiang Province in Northern China [29,30]. Among the several risk factors for liver cancer, chronic hepatitis B and C virus infections are distinctive because they are highly prevalent in East Asia but not in Western countries [31,32]. ESCC is preceded by recurrent exposure to very hot substances as a significant risk to the lining of the esophagus due to repeated thermal injuries [33,34]. A higher incidence of congenital biliary cysts and anomalous pancreaticobiliary duct junctions in East Asia increases the risk for gallbladder cancer through chronic inflammation [35].

#### 1.1.5. Transition to Microbiota Focus

While these risk factors lay the groundwork in large accounts for understanding EC, recent studies have discovered a notable role of the microbiota in cancer genesis, especially concerning EC. Microorganisms also live in the human mouth, a complex environment with places for bacteria, among archaea, fungi, and viruses. For example, the bacterial communities in the oral cavity are much less diverse and variable than at other body sites, like on the skin or within the gut. Many studies have shown that oral bacterium and gastrointestinal tumors have a tight relationship [36]. These microorganisms can incite cancer development through their ability to produce carcinogenic compounds, causing constant inflammation and alteration in host cell metabolism.

Environmental factors such as smoking and alcohol consumption, which are known to contribute to chronic inflammation and epithelial cell transformation, are also known to affect the composition of the oral microbiota [37]. These factors, by increasing the risk for a dysbiotic state in which harmful bacteria can colonize and beneficial species are lost, lead to disease and even up to cancer [38]. This underlines the critical role of chronic inflammation in cancer genesis, urging us to be more aware and cautious.

New studies have indicated that changes in the esophageal microbiota order ratio to Gram-negative bacteria will cause inflammation, TLR-4 activation with LPS, and increased reflux from the relaxation of the nitric oxide synthase (NOS)-mediated sphincter muscle [39]. However, the direct contribution of esophageal microbiota to EC is still poorly understood despite the emerging insight into how these microbes interact with cancer.

The human microbiome has been established to play a crucial role in the pathogenesis of human cancers. Recent technological improvements, such as metagenomic DNA sequencing, provide higher-resolution insights, opening up potential for using microbiota as another means to treat numerous diseases, including cancer. However, to fully harness this potential, it is important to study more about the interaction between microbiome and the immune system. This will provide novel strategies for EC therapy with relevant biomarkers, equipping us in the exciting frontier of cancer research [40].

This review highlights the recent research indicating that EC and, more specifically, ESCC are closely linked to chronic inflammation and the composition of gut or esophageal microbiota. We further discuss several microbiota-mediated pathways as potential targets of EC therapy and explore strategies to enhance prevention, early diagnosis/screening, and treatments for this highly lethal disease.

## 2. Epidemiology Findings

Recent studies have begun to characterize the significant microbial variations linked to ESCC. Such modifications expose the novel biomarkers useful for early diagnostics, the prediction of disease outcomes, and the response to therapeutic interventions. Different researchers have demonstrated that the over-abundance of specific microbes in ESCC tissues compared to the non-tumor or healthy controls, respectively, suggests a specific role of the microbiota during the pathogenesis of ESCC (Table 1).

Numerous subsequent studies have consistently found a higher abundance of pathogenic organisms in the ESCC tissues relative to the non-tumor tissues or the healthy controls. *F. nucleatum* has been found to be more frequently enriched in ESCC tissues, with several studies by Jiang et al. (2021), Shao et al. (2019), and Yamamura et al. (2016). *F. nucleatum* abundance leads to the progression of tumor stages and worse clinical outcomes. This indicates a possible future biomarker for ESCC development and prognosis [41,42,47].

*P. gingivalis* is one of the critical species in ESCC tissues. For example, Gao et al. (2016) demonstrated this bacterium in 61% of cancerous tissues and validated a significant association with ESCC development [44]. The presence of *P. gingivalis* was closely associated with the differentiation status, metastasis, and overall survival rates in ESCC patients, suggesting that it may serve as a potential clinical target for ESCC treatment.

Recent studies have also investigated the microbial diversity and composition in ESCC tissues, showing a modest reduction in general bacterial richness. Jiang et al. (2021) and Li et al. (2020) found that all the alpha diversity indices were decreased in tissues of ESCC, suggesting a reduction in microbial taxa richness; moreover, dramatic differences were observed in specific microbial phyla and genera community [42,43]. ESCC tissues generally showed upregulated *Fusobacteria*, *Bacteroidetes*, and *Firmicutes*, while *Actinobacteria* and *Proteobacteria* were less than in the healthy controls [43,45].

Furthermore, additional studies of the associations of specific microbial genera in the clinicopathological phenotypes of ESCC have provided more proof for a relationship between this disorder and microbiota. For example, the tissue samples from ESCC patients compared to the normal esophagus showed that the *Streptococcus* and *Prevotella* genera were enriched in ESCC and have been statistically associated as an event seen along disease progression [42,49]. The presence of *Proteus* in ESCC and the change in *Firmicutes* and *Bacteroides* abundance among different the morphological types of ESCC implies that microbial composition may affect tumor traits [48].

Studies on the prognostic value of these microorganisms have been performed as well. The presence of *F. nucleatum* showed a statistically significant association with a shorter survival time, indicating that it might be a prognostic biomarker [47]. Kovaleva et al. (2021) concluded that *Staphylococcus* had a positive relationship with inflammatory tumor infiltrates and the resulting prognostic value in ESCC through the two sets of patients studied [50].

The relationship between the microbiota and the tumor microenvironment (TME) in ESCC has been investigated in-depth. For example, Lin et al. (2022) revealed that the microbial co-occurrence networks were significantly denser and, by far, more complicated in tumor-adjacent tissues compared to tumor tissues, with differentially abundant microbiota being predominantly linked, in the tumor-adjacent tissues, to the signaling pathways involved in carcinogenesis [46].

Functional studies have also uncovered the specific roles of certain microbiota in ESCC. For instance, Yang et al. (2021) found that ESCC-associated microbiota exhibited altered nitrate and nitrite reductase activity, suggesting that these functional changes may contribute to the development of esophageal cancer [45].

In conclusion, the reviewed studies show significant changes to the esophageal microbiota associated with ESCC. The organisms observed to have undergone changes include *F. nucleatum* and *P. gingivalis*. Some of these changes might be possible biomarkers for ESCC diagnosis and prognosis. The decreased microbiota diversity and specific functional changes have also been detailed. This can help in comprehending how the microbiota causes esophagus cancer. Understanding the probable diagnoses and the mechanisms linking the various changes is crucial since it sluices epidemiologic studies. A total comprehension of microbiota concerning its carcinogenic role may help develop microbiota-based therapy for ESCC.

## 3. Chronic Inflammation

### 3.1. Immune Regulation by Microbiota

The esophageal microbiota regulates inflammation and immune responses through interactions with the mucosal immune cells (Table 2).

Dysbiosis disrupts this balance, leading to chronic inflammation and oncogenesis. Key pathogenic bacteria such as *P. gingivalis* and *F. nucleatum* significantly contribute to cancer development by activating inflammatory pathways, including nuclear factor kappa-light-chain-enhancer of activated B cells (NF-κB), extracellular signal-related kinases 1 and 2–E26 transformation-specific sequence 1 (ERK1/2–Ets1), and nucleotide-binding oligomerization domain-containing protein 1/receptor-interacting serine/threonine-protein kinase 2/NLR family pyrin domain containing 3 (NOD1/RIPK2/NLRP3) inflammasome [53,54,55,61]. These interactions lead to the production of pro-inflammatory cytokines such as interleukins IL-1β, IL-6, IL-8, and tumor necrosis factor-α (TNF-α). Microbial metabolites, such as SCFAs from beneficial microbiota, modulate regulatory T (Treg) cell function by inhibiting histone deacetylases (HDACs) and thus impacting immune homeostasis [58,59,60]. Dysbiosis, characterized by harmful bacteria prevalence and reduced SCFA-producing bacteria, creates an inflammatory microenvironment conducive to ESCC. This chronic inflammation disrupts the epithelial barrier and reprograms the immune cells within the TME, promoting tumor growth, metastasis, and therapy resistance [65] (Figure 2).

### 3.2. Activation of Inflammatory and Signaling Pathways

ESCC development has been linked to activating inflammatory pathways and producing cytokines, chemokines, and ROS. The LPS of Gram-negative bacteria, such as *P. gingivalis* and *F. nucleatum*, binds to TLR-4 on immune cells and activates the MyD88-dependent pathway, leading to the activation of NF-κB and the production of pro-inflammatory cytokines IL-1β, IL-6, IL-8, TNF-α in ESCC [62,66,67]. Additionally, *P. gingivalis* activates the ERK1/2–Ets1 and PAR2/NF-κB pathways, while the activation of the NOD1/RIPK2/NF-κB and NLRP3 inflammasome pathways acts as downstream effectors for *F. nucleatum*. Inflammatory pathways that become activated destabilize the epithelial barrier and activate the host’s DNA damage and pro-oncogenic signals, triggering the carcinogenesis process [56]. More so, the persistency of NF-kB activation is promoted by the continuous exposure to microbial products for the host immune cells, causing increased inflammation, cell proliferation, protection of cells from apoptosis, and inducing angiogenesis through vascular endothelial growth factor (VEGF) expression [57]. Chronic inflammation resulting from pathogenic microbiota increases the levels of pro-inflammatory cytokines, which contribute to ESCC development. TLRs are activated by *Escherichia coli*, the most prevalent organism in both BE and EAC, suggesting that this organism has an early role in carcinogenesis [63]. In addition, *A. actinomycetemcomitans* generates factors that exaggerate inflammation and cancer hazard [64].

### 3.3. TME and Immune Reprogramming

The TME in EC is a complex biological environment that contains the immune cells, endothelial cells, cancer-associated fibroblasts, adipocytes, extracellular matrix proteins, and secretory factors such as chemokines, cytokines, and growth factors. The TME is infiltrated with cells programmed to perform immunosuppressive functions, including tumor-associated macrophages, myeloid-derived suppressor cells, and Treg cells [68,69]. Pathogenic bacteria, notably *P. gingivalis* and *F. nucleatum* modulate the TME by activating chemokines, growth factors, and cytokine production from tumor cells [63,70,71]. In the progression of EC, persistent chronic inflammation promoted via dysbiosis and pathogenic bacteria is crucial. Continuous stimulation of the pro-inflammatory pathways, especially the signal transducer and activator of transcription 3 (STAT3)- and NF-kB-dependent cascades, maintains a prototypic inflamed tumor microenvironment, independently driving the initial steps that lead to oncogenesis. The IL-6 binds with its receptor (IL-6Rα) using the signaling of the STAT3 pathway and is related to poor prognosis in EC [72]. The potential therapeutic value of treatment with selective inhibitors of STAT3 is being preclinically explored. *P. gingivalis* stimulates the ERK1/2–Ets1 and PAR2/NF-κB pathways, whereas *F. nucleatum* activates NOD signaling through RIPK2-dependent NF-κB and inflammasomes to induce cytokines that drive chronic inflammatory immune-suppressive activity [53,61,73]. *A. actinomycetemcomitans* produces inflammatory and immunosuppressive virulence that helps with TME modulation [64]. Targeting pathogenic bacteria and signaling pathways in the TME shows promise as a therapeutic approach for EC.

## 4. Microbial Dysbiosis 

The development of ESCC is related to microbial dysbiosis (Table 3), disrupted host–microbial interactions that alter the regular composition of the microbial community. An imbalance in the microbial composition is linked with ESCC; however, multiple risk factors, such as hormonal imbalances, dietary compounds, toxins, and antibiotics, contribute to dysbiosis. Microbial dysbiosis promotes immune regulation disruption and the inclusion of chronic inflammation in causative factors associated with ESCC and oncogenesis. Reduced diversity of esophageal microbiota has been observed in patients with ESCC, including the reduced proportion of the beneficial *Streptococcus* species and an increased number of pathogenic bacteria such as *P. gingivalis* and *F. nucleatum* that contribute to a tumor-promoting microenvironment [43,44,46]. Pathogenic bacteria are important as ESCC correlates with the progression of the disease, poorer prognosis, and a severe response to chemotherapy. *F. nucleatum* leads to shorter survival and increased tumor behavior by activating chemokines, such as CCL20 [47]. *P. gingivalis* is also related to ESCC severity with a decreased survival rate such that these bacteria may serve as biomarkers for ESCC targeting [44].

The adaptive immune system functions with the participation of the commensal bacteria from the human body. The dysbiosis of such bacteria disrupts the interaction and thus leads to immune deregulation and cancer progression. Some pathogens, such as *P. gingivalis* and *F. nucleatum*, avoid the immune response and induce a persistent inflammation and carcinogenesis by activating the NF-κB, ERK1/2–Ets1, and NOD1/RIPK2/NLRP3 inflammasome pathways. Thus, these pathways provoke the secretion of pro-inflammatory cytokines and the onset of the barrier pathogen-induced disease [53,54,55,61]. At the same time, diet and many external factors profoundly impact microbial dysbiosis. For instance, it was clear that a high-fiber diet strengthens the beneficial bacteria *Firmicutes*. However, the absence of fiber in the diet, as well as alcohol consumption and tobacco usage, activate the harmful Gram-negative bacteria, for example, *Prevotella*, *Neisseria*, and *Eikenella*. The secreted endotoxins target the epithelial cells, initiating their inflammation and carcinogenesis [77,82,83,84].

Microbial dysbiosis is obvious in GERD and BE, where increasing Gram-negative bacteria intensify mucosal damage and inflammation, driving the development from benign to malignant tissue [85]. Proton pump inhibitors reduce stomach acid production, which alters the esophageal microbiota by increasing gastric pH and decreasing exposure to acid, which can disrupt the balance of beneficial and harmful bacteria [86].

Motility disorders, such as achalasia, further add to the dysbiosis state because food stasis provides a playground for bacteria growth by reducing esophageal clearance. The range of potential pathogens is much broader in cancer patients and can include *Prophyromonas*, *Prevotella*, or *Fusobacterium*, which are abundantly common among achalasia patients and may act to promote inflammation and carcinogenesis [87,88].

Recent evidence has begun to explore the particular presence of pathogens, such as *P. gingivalis* and *F. nucleatum*, in ESCC. In order to thrive and continue growing, these bacteria work together to suppress chronic inflammation while promoting immune evasion, which allows for tumor survival and progression. Understanding these interactions can help develop targeted therapeutic strategies to manage dysbiosis and reduce the risk of EC.

## 5. Production of Carcinogenic Metabolites

The production of carcinogenic metabolites by various bacteria significantly contributes to the development and progression of EC (Table 4). Anaerobic bacteria such as *Bacteroides*, *Clostridium*, *Faecalibacterium*, and *Ruminococcus* produce SCFAs like butyrate, acetate, and propionate through the fermentation of dietary fibers [89,90,91,92]. These SCFAs are essential for maintaining intestinal barrier integrity and exerting anti-inflammatory effects. However, in EC patients, a reduction in SCFA production weakens the intestinal barrier and contributes to a pro-inflammatory environment, promoting carcinogenesis [93,94]. SCFAs inhibit histone deacetylases (HDACs), influencing Treg cell function and reducing inflammation; their deficiency thus fosters tumor development [58] (Figure 3).

Alcohol is metabolized into acetaldehyde by bacteria such as *Neisseria*, *Streptococcus*, and the fungus *Candida*, which have high alcohol dehydrogenase (ADH) activity [95]. Acetaldehyde is a toxic and carcinogenic metabolite that causes DNA damage, mutagenesis, and disrupts the gut microbiota, significantly increasing the risk of EC through chronic exposure [105].

The production of ROS during microbiota-induced inflammation is another mechanism by which cancer development occurs. Microbes like *P. gingivalis*, *H. pylori*, and *E. coli* produce ROS to infect the host cells. Hence, *P. gingivalis* secretes nucleoside diphosphate kinase (NDK) that regulates ATP-mediated ROS release. ROS release does significant DNA damage and promotes the activation of transcription factors that cause inflammation and facilitate the progression of cancer [96,97,106,107]. In addition, reactive nitrogen species (RNS) are products of some microbes that promote the activation of nitrosative stress, which causes DNA damage and contributes to cancer progression. RNS are products of *S. oralis*, *S. mitis*, *S. sanguinis*, *S. gordonii*, *L. fermentum*, *L. jensenii*, *L. acidophilus*, and *B. adolescentis* [98,99]. Matrix metalloproteinases (MMPs) are essential for the movement of cancer cells from the original site to other sites. *P. gingivalis* produces gingipains that activate MMP-9, while *F. nucleatum* stimulates p38 signaling and leads to the secretion of MMP-9 and -13 [55,100]. Hence, one of the primary roles of MMPs is to facilitate cancer metastasis through the degradation of the extracellular matrix [108].

Hydrogen sulfide (H_2_S) is a volatile sulfur compound produced by oral bacteria, such as *P. gingivalis*, *P. intermedia*, *A. actinomycetemcomitans*, and *F. nucleatum*, which is genotoxic and induces genomic instability. Moreover, the accumulation of mutations stimulated by H_2_S advances tumor growth and dissemination by activating several signaling pathways [101,102]. Another genus of aciduric bacteria, *Lactobacillus*, *Lactococcus*, *Bifidobacterium*, *Streptococcus*, *Leuconostoc*, and *Pediococcus*, compel the fermentation of glucose to lactic acid. Furthermore, the excessive production of lactic acid initiates an acidic and hypoxic atmosphere of the tumor, suppresses immune responses, and enhances metastatic efficiency, facilitating tumor progression [103,109,110].

*E. coli* secretes colibactin, a metabolic genetic toxic substance with a potent carcinogenic effect, causing the emergence of DNA double-strand breaks and genomic instability, which contributes highly to carcinogenesis [104,111].

In conclusion, the microbial genera secreting carcinogenic metabolites, especially SCFAs and acetaldehyde, ROS, RNS, MMPs, H_2_S, and lactic acid, are closely associated with EC. Metabolites of microbial origin form a pro-inflammatory microenvironment that also damages DNA and indirectly stimulates the growth and invasion processes characteristic of oncogenesis/metastasis via the suppression of an immune response. This understanding, therefore, opens up potential pharmacological targets for controlling and reducing the risk of EC due to microbial dysbiosis and metabolite production.

## 6. Direct Interaction with Epithelial Cells

The microbiota are directly involved in interactions with the esophageal epithelial cells, playing a critical role in the development and progression of EC (Table 5). Pathogenic bacteria such as *P. gingivalis* and *F. nucleatum* target epithelial cells through multiple mechanisms that enhance carcinogenesis. Disrupting the balance between commensal and pathogenic bacteria in the esophageal mucosa can lead to diseases driven by harmful bacterial strains. For instance, *P. gingivalis* and *F. nucleatum* utilize specific adhesion molecules to bind to epithelial cells, leading to the activation of intracellular signaling pathways. *F. nucleatum*, through its FadA adhesin, binds to E-cadherin on the epithelial cells. It disrupts the epithelial barrier and promotes cancer progression by activating β-catenin signaling, which enhances cancer cell proliferation and migration [112,113].

In addition to adhesion, these bacteria can invade epithelial cells and deliver virulence factors that modulate cell signaling pathways. *P. gingivalis* triggers the ERK1/2–Ets1 and PAR2/NF-κB axes, while *F. nucleatum* activates the NOD1/RIPK2 pathway, potentiating the proliferation and migration of ESCC cells [55,61]. These direct bacterial interactions contribute to esophageal diseases such as GERD and BE, where the presence of bacteria like *Campylobacter*, *Leptotrichia*, *Fusobacterium*, *Rothia*, and *Capnocytophaga* is enriched [79,119]. These interactions with esophageal epithelial cells lead to chronic inflammation, tissue damage, and epithelial transformation, creating a pro-inflammatory environment conducive to cancer development. The long-term activation of TLRs, particularly TLR-2 and TLR-4, by bacterial components promotes chronic inflammation in BE and EAC, facilitating tumor progression [122,123].

Bacteria can also induce malignant transformation by modulating cellular processes such as epithelial–mesenchymal transition and apoptosis. *P. gingivalis* and *F. nucleatum* are known to induce EMT, predisposing cells to malignant transformation [124]. *P. gingivalis* inhibits apoptosis through the JAK1/AKT/STAT3 pathway, reducing apoptotic activity by increasing the B-cell lymphoma-2 (BCL-2) to BCL-2-associated X-protein (BAX) ratio [114,125,126,127]. Meanwhile, *F. nucleatum* activates TLR-4, leading to the upregulation of autophagy and downregulation of apoptosis, creating a chemoresistant phenotype. It also binds to E-cadherin, promoting β-catenin signaling and further enhancing cancer cell proliferation [70,118].

In addition, certain bacteria release outer membrane vesicles (OMVs) that carry virulence factors such as LPS directly into the cytoplasm of epithelial cells. For example, *P. gingivalis* releases OMVs that fuse with the epithelial cell membrane, activating TLR-4 and triggering NF-κB signaling, which results in the production of pro-inflammatory cytokines, such as IL-1β, IL-6, and TNF-α [128,129]. These cytokines promote a pro-inflammatory environment that contributes to tumor development.

Furthermore, virulence factors produced by *A. actinomycetemcomitans* and other bacteria enhance neoplastic transformation by directly interacting with epithelial cells [64]. *F. nucleatum*, in particular, establishes a strong adherence to host cells via FadA adhesin, significantly increasing the risk of tumor development [130]. Prolonged *P. gingivalis* infection can also lead to the acquisition of cancer stem cell properties in non-neoplastic cells by modulating cyclin-dependent kinase (CDK) function, promoting uncontrolled cellular proliferation [115].

These data underscore the pivotal role of direct bacterial interactions with esophageal epithelial cells in modulating inflammatory responses and cellular processes that contribute to carcinogenesis. Understanding these interactions provides critical insight into the pathways driving EC development and highlights the potential therapeutic targets for controlling and reducing EC risk.

## 7. Epigenetic Modifications 

Epigenetic modifications refer to the heritable changes in gene expression that occur without any changes to the DNA sequence. These modifications significantly affect the course of EC development and are mainly affected by microbiota alterations (Table 6). Microbiota produces SCFA substances, such as butyrate, that influence HDACs. The inhibition then affects the Treg cells’ function, altering the immune reaction and local inflammation. An optimal proinflammatory milieu will create a favorable condition to initiate carcinogenesis [58,59,60]. In BE and EAC, the activation of TLR-4 can directly affect the expression of cyclooxygenase-2 (COX-2), which occurs through NF-κB-independent mechanisms, such as mitogen- and stress-activated kinase (MSK) and mitogen-activated protein kinase (MAPK), that potentially involve altered epigenetic factors [123,131] (Figure 4).

*P. gingivalis* infection upregulates miR-194 and Akt and downregulates grainy head-like transcription factor 3 (GRHL3) and phosphatase and tensin homolog (PTEN), promoting esophageal tumors’ pro-proliferative and pro-migratory phenotype [132]. Thus, *P. gingivalis* modulates miRNA expression in cancer progression through epigenetic regulation [137].

Numerous genetic changes were characterized following the interaction of the oral microbiota with the epithelial cells. Specifically, mRNAs, miRNAs, and long non-coding RNAs (LncRNAs) were differentially expressed [135,136]. When P53 is downregulated as a tumor suppressor gene, several changes occur in the epithelial cells, which is characteristic of malignant transformation [138]. These changes promote EC development by interfering with cellular regulation.

Additionally, *F. nucleatum* shifts the macrophage infiltration and methylation status of the cyclin-dependent kinase inhibitor 2A (CDKN2A) promoter in cancerous lesions [133]. This interaction alters gene expression profiles, leading to malignant transformation by repressing tumor suppressor genes and inducing oncogenes. The transcriptional activation of oncogenes such as C-myc and cyclin D1 by *F. nucleatum* is mediated via the upregulation of β-catenin signaling, suggesting a direct mechanistic link between microbial infection with subsequent epigenetic modifications leading to cancer induction [70,134].

To summarize, microbiota-induced epigenetic variation substantially contributes to the development of EC. Such changes involve the inhibition of HDAC by SCFAs, activation of TLR-4 on COX-2 expression, modulation of miRNA by *P. gingivalis*, changes in the genetic coding and DNA methylation patterns by *F. nucleatum*, and signaling in the EC with the other oncogenic pathways, such as β-catenin signaling. Indeed, all these processes cooperate to perform crucial roles in promoting EC development by facilitating the bidirectional relationship between the microbiota and host cells.

## 8. Interaction with GERD

GERD significantly impacts the esophageal microbiome, leading to dysbiosis that favors chronic inflammation and carcinogenesis (Table 7). Patients with GERD exhibit an increased proportion of Gram-negative bacteria, including *Veillonella*, *Prevotella*, *Neisseria*, and *Fusobacterium*, along with a reduction in Gram-positive bacteria [139,140]. This results in a higher Gram-negative/Gram-positive ratio, which is closely associated with reflux esophagitis and BE—a premalignant condition that can progress to EAC [141].

Microbiota changes in GERD contribute to chronic inflammation through the release of LPS from Gram-negative bacteria, which activate TLR-4 on the epithelial or immune cells. This interaction triggers the NF-κB signaling pathway, leading to the production of pro-inflammatory cytokines such as IL-1β, IL-6, IL-8, and TNF-α [142,153,154]. The resulting chronic inflammation promotes epithelial injury and transformation, predisposing patients to cancer development. *F. nucleatum* plays a key role in this process, enhancing cytokine production and activating NF-κB, while the *Campylobacter* species, often enriched in GERD and BE tissues, upregulate the cytokines linked to carcinogenesis.

Chronic inflammation driven by GERD not only disrupts the esophageal microbiome but also promotes the production of ROS, further contributing to DNA damage and carcinogenesis. Long-term exposure to gastric acid and bile salts results in persistent inflammation and tissue injury, facilitating the transition from metaplasia in BE to dysplasia and ultimately to EAC. Sustained activation of TLR-2 and TLR-4 by the microbial components exacerbates this process, mediating inflammation, cell proliferation, and carcinogenesis.

Age-related microbiota shifts and inflammation further increase cancer risk in GERD patients [155]. For example, studies highlight an increased abundance of *Streptococcus* in elderly individuals, which may exacerbate chronic inflammation and carcinogenesis. Additionally, *Escherichia coli* upregulates TLRs, particularly TLR-4, stimulating early carcinogenic events through the increased production of pro-inflammatory cytokines.

The following bacterial species have been implicated in the microbiota alterations seen in GERD and BE, promoting chronic inflammation and epithelial transformation:*Campylobacter*: It is overrepresented in GERD and BE patients, causing the induction of chronic inflammation and mucosa alteration that might contribute to EAC emergence [145].*F. nucleatum*: It binds to or invades epithelial cells, modulates the immune response, and promotes inflammation, which enhances the progression from BE to EAC through TLR activation [146].*Prevotella*: It is overrepresented in GERD, a precursor to BE and EAC, and it may facilitate chronic inflammation and mucosal damage [147].*S. anginosus*: It is linked to GERD and BE and can create local chronic inflammation and the epithelial cell perturbations associated with progression to the development of EAC [71,148,156].*Leptotrichia*: It is overrepresented in patients with GERD and BE, promoting ongoing inflammation and metaplasia of the epithelial monolayer and leading to carcinogenesis [149,150].*Rothia*: It is increased in GERD and BE patients, causing chronic inflammation and mucosal damage that promotes progression to EAC [151].*Capnocytophaga*: It tends to be enriched in GERD and BE patients, mechanistically promoting chronic inflammation and esophageal mucosal changes, thereby creating conditions conducive to EAC [152].

The chronic inflammatory state driven by GERD, coupled with microbial dysbiosis, plays a pivotal role in promoting esophageal carcinogenesis. The interaction between microbial components, such as LPS, and host receptors, like TLR-4, triggers inflammatory pathways [141,157,158]. Simultaneously, the persistent production of ROS exacerbates DNA damage and tumor progression [159]. Additionally, high-fat diets, often associated with obesity, further contribute to GERD, BE, and EAC risk by reducing microbial diversity and promoting an inflammatory, cancer-prone environment. Obesity, a major risk factor for GERD, increases the prevalence of BE and EAC and may act synergistically with GERD-induced microbial changes to elevate cancer risk [160,161].

*H. pylori* also plays a critical role in this process, particularly in inducing chronic gastritis, which impacts gastric acid secretion and may promote the progression of GERD to BE and EAC [144,162]. The complex interplays among aging, diet, obesity, microbial dysbiosis, and GERD underscores the multifactorial pathways contributing to esophageal carcinogenesis.

## 9. Metabolic Changes and EC

The abiotic factors driving the gut microbiota-related metabolic changes accessible for development and progression are metabolite production, dysbiosis, dietary habits, and obesity. Table 8 presents the specific bacteria associated with these changes that promote carcinogenesis through various mechanisms. The metabolites function as SCFAs, bile acids, and branched-chain amino acids typically released by microorganisms, which help regulate immune function and digestion [89,90,91,92]. For instance, bacteria such as *Bacteroides*, *Clostridium*, *Faecalibacterium***,** and *Ruminococcus* produce SCFAs that support immune function and gut homeostasis. However, a reduction in the levels or activity of these beneficial bacteria can lead to decreased SCFA production, creating a pro-inflammatory environment that contributes to an increased risk of cancer [163].

Dysbiosis is the imbalance of microbiota due to the environment, stress, or low immune function. It disrupts the microbiota’s average balance, resulting in various metabolic disorders, obesity, insulin resistance, and chronic inflammation. The specific reduced diversity of oral microbiota is highly associated with ESCC. *H. pylori* and *Campylobacter* are among the bacteria that cause these metabolic diseases [46,164]. A Western diet, which is high in fat and low in fiber, is associated with systemic low-grade inflammation and an increased load of various metabolic diseases. Furthermore, this diet disrupts the gut microbiome and bile acid metabolism, leading to the formation of BE and EAC [165,166]. Obesity, which is also associated with the Western diet, is additionally considered a risk factor for GERD, BE, and EAC.

Lactic acid-producing bacteria including *Lactobacillus*, *Streptococcus*, *Bifidobacterium*, and *Leuconostoc*, also facilitates the disease progression. They can create a low pH and hypoxic microenvironment conducive for tumor metastasis where they induce the Warburg effect [80] that supports cancer cell survival and proliferation, thereby promoting EC progression [167]. In addition, it has been reported that *F. nucleatum* can produce LPS that activates β-catenin signaling, thus enhancing oncogene expression (C-myc and cyclin D1) and promoting cancer cell proliferation [134].

*P. gingivalis* similarly modifies adenosine triphosphate/P2X purinoceptor 7 (ATP/P2X7) signaling, which affects ROS and antioxidant response and therefore contributes to cancer development through ROS-induced DNA damage and inflammation [106]. *Streptococci* and *Candida* also metabolize alcohol to acetaldehyde, a toxic metabolite responsible for the development of DNA damage, leading to the risk of carcinogenesis [95].

In summary, metabolic changes driven by specific bacteria and environmental factors significantly contribute to the development and progression of EC. These changes include the production of functional metabolites, dysbiosis, dietary influences, lactate metabolism, LPS and cytokine production, and alcohol metabolism to acetaldehyde.

## 10. Angiogenesis

Angiogenesis, the generation of new blood vessels from pre-existing vasculatures, is fundamental to tumor progression as it supplies growth and metastasizing with essential nutrients and oxygen. In EC, the inflammatory microenvironment triggered by several bacteria promotes angiogenesis to a final extent through multiple pathways (Table 9).

ROS generated within the inflammatory microenvironment are implicated in cancer initiation and progression by inducing mutagenesis and enhancing angiogenesis. *H. pylori* and other bacteria may increase ROS production thus activating angiogenesis and contributing to cancer development. In addition, this mechanism has been well documented in gastric cancer with relevance to EC (Figure 5). Stabilizing hypoxia-inducible factor 1-alpha (HIF-1α) in the core of tumors upregulates the surrounding pro-angiogenic factor such as VEGF [175,176]. Serum levels of VEGF have been documented to increase in association with disease progression, with raised levels correlating with increased tumor burden and predicting poor prognosis in the case of ESCC [177]. At the same time, *H. pylori* may participate in this relation as it contributes to hypoxia, stabilizing HIF-1α [169].

Most importantly, IL-8 was determined to support angiogenesis and the proliferation and migration of cancer cells [178]. Angiogenesis is considered a hallmark of EAC development as it ensures the growth of the tumor by creating a source of nutrients. *H. pylori* and *F. nucleatum* upregulate IL-8 levels in an effort to foster angiogenesis [113,142,179].

Inflammation changes the TME considerably and can aggravate the development of a disease by triggering the process of angiogenesis. Macrophages and dendritic cells found in the tissues affected by EAC begin to produce VEGF and MMPs to support angiogenesis and make a particular tumor more invasive [180,181]. At the same time, *P gingivalis* and *F. nucleatum* are also involved in these processes, affecting an inflammatory response and cytokine generation [170]. IL-1β is produced after infection and is known to activate endothelial cells and trigger VEGF production and other pro-angiogenic factors. As a result, the TME becomes inflammatory and more suitable for angiogenesis and tumor progression [182,183]. Bacteria such as *H. pylori* and *F. nucleatum* significantly impact this cytokine, exacerbating the processes of IL-1β production [184].

TNF-α enhances the expression of many angiogenic factors, such as IL-8, VEGF, and basic fibroblast growth factor (bFGF), which promote angiogenesis [185,186]. In addition, bacteria such as *P. gingivalis* and *F. nucleatum* increase the levels of TNF-α and thus intensify the process of angiogenesis [66,171]. Moreover, H_2_S, produced by oral bacteria and *P. gingivalis*, affects the development and spread of the tumor and the activation of the growth and migration of different signals, which are invasive pathways, promoting the process of tumor angiogenesis [101,187]. Consequently, H_2_S helps to produce the microenvironment, which is favorable for building new blood vessels.

To sum up, it is evident that the process of EC angiogenesis is significantly affected by some bacteria impacting the inflamed reactions and generating the pro-angiogenic factors. These include ROS formation, HIF-1α stabilization, cytokine and metabolite production, and H_2_S activity. All the effects of bacteria on the process of angiogenesis may be used for the further research of the possible tasks of therapy target for EC.

## 11. Future Directions

Microbiome involvement in the development and progression of EC has opened up new perspectives for research and treatment. Additional studies are indispensable to elucidate more intricate mechanisms of how particular bacteria, such as *F. nucleatum* and *P. gingivalis*, promote carcinogenesis. A combination of in vitro, in vivo, and clinical studies is necessary to reach this level. In vitro studies on the co-culture systems of esophageal epithelial cells with these bacteria will provide insight into the direct interactions, cellular responses, and pathway activations. Additionally, advanced tools such as RNA sequencing and metabolomics will uncover the mechanism of altered gene expression or metabolic changes induced by these bacteria; meanwhile, cell proliferation activity assays (apoptosis, migration capacity) would further establish their involvement in cancer development. The construction and usage of animal model systems, like xenografts or those with genetically altered mice for microbial-induced carcinogenesis, will be essential to establish the in vivo significance of cancer. Specifically engineered “germ-free” mice with bacterial populations of interest could prove to be pivotal indirect evidence for the microbiota’s role in tumor development, allowing for additional studies on inflammation and immune responses and sensitivity to potential therapeutic intervention opportunities.

Correlating microbiome analysis with the clinical data of EC, GERD, and healthy individuals will be vital in identifying the potential significance of these changes. Methods like 16S rRNA sequencing, metagenomics, and metatranscriptomics will be invaluable to profiling microbial communities and revealing the architectural features of microorganisms that determine the function. Our next step is correlating these findings with clinical outcomes, which may provide the candidate biomarkers for early detection and prognosis. It will be necessary to validate candidate biomarkers found in primary studies in larger patient cohorts and develop sensitive and specific assays for their detection through a biological sample that could easily standardized, like blood, saliva, or biopsy tissue, to help them become clinically useful.

Consequently, machine learning algorithms that combine microbiome and metabolomics information with clinical variables can be implemented to build predictive models for the outcomes of EC patients. Moreover, drug screening for small molecule inhibitors of bacterial virulence factors and cancer-promoting pathways will open a new door to developing antisense-based therapeutics. Antibiotics, probiotics, and fecal microbiota transplantation (FMT) should be further investigated for their efficacy in manipulating the microbiome during cancer progression. The potential of these immunotherapeutic strategies through immune checkpoint inhibitors and vaccines against *F. nucleatum* and *P. gingivalis* is also discussed to improve anti-cancer immunity.

## 12. Conclusions

In this review, we discuss the mechanistic role of the microbiota in EC development and progression. Our review emphasizes the role of a complex network within EC pathogenesis involving not only these pathogens but also further enhanced by other microbial metabolites and is driven, in part, by chronic inflammation and immune modulation. Targeting the microbiota opens new avenues for diagnosis, prognosis, and therapeutic development and, more importantly, their ability to drive these associations. The findings suggest that future studies should include more comprehensive microbiome profiling, investigating these mechanisms with a focus on gut microbiota-targeted, personalized medicine to identify the candidate strategies for preventing and treating this highly aggressive form of human cancer.

## Figures and Tables

**Figure 1 cancers-16-03305-f001:**
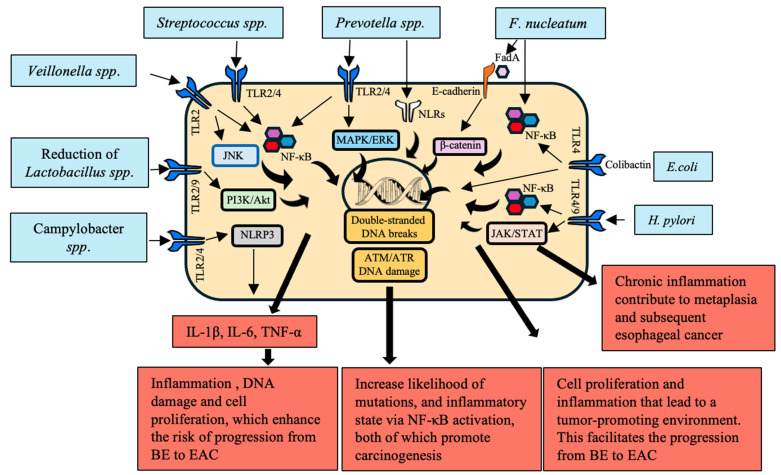
Reflux-induced changes in the esophagus and their tumorigenic outcomes mediated by microbial interactions. This figure illustrates the key microbial species involved in the pathogenesis of gastroesophageal reflux disease (GERD) and Barrett’s Esophagus (BE), highlighting their interactions with esophageal epithelial cells, specific receptors, signaling pathways, and the resultant tumorigenic outcomes. The dominant microbiota implicated in these processes include *Streptococcus* spp., *Prevotella* spp., *Fusobacterium nucleatum*, *Escherichia coli*, *Veillonella* spp., *Lactobacillus* spp., *Campylobacter* spp., and *Helicobacter pylori*. These microbes activate host receptors such as toll-like receptors (TLR2, TLR4), NOD-like receptors (NLRs), and E-cadherin, triggering critical signaling cascades including NF-κB, β-catenin, MAPK/ERK, and ATM/ATR pathways. The activation of these pathways results in the production of pro-inflammatory cytokines, such as IL-1β, TNF-α, IL-6, and IL-8, which promote chronic inflammation, epithelial barrier disruption, and DNA damage. These processes lead to increased epithelial cell proliferation, inhibition of apoptosis, and the development of a tumor-promoting microenvironment that drives the transition from GERD to BE and, ultimately, to esophageal adenocarcinoma (EAC). The figure also emphasizes the reduction in protective microbiota like *Lactobacillus* spp., which weakens the mucosal barrier and further enhances the inflammatory milieu. In particular, *F. nucleatum* activates β-catenin and NF-κB signaling, promoting immune evasion and carcinogenesis, while *E. coli* induces DNA damage through colibactin production, triggering the ATM/ATR DNA damage response pathway. *Prevotella* spp. and *Veillonella* spp. contribute to biofilm formation, exacerbating inflammation and tissue damage through TLR2 and NLR activation. The direct involvement of *Campylobacter* spp. and its production of carcinogenic nitrosamines, along with the interplay of these microbial species and their inflammatory effects, underpin the progression toward malignancy.

**Figure 2 cancers-16-03305-f002:**
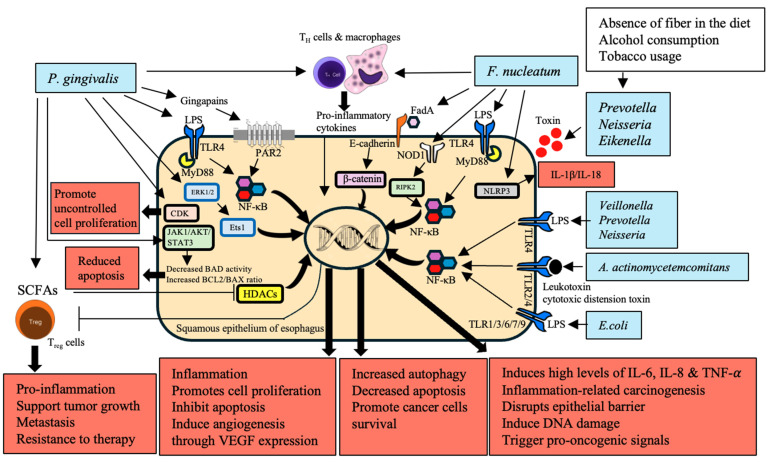
Role of inflammation, dysbiosis, and microbial interactions with epithelial cells in the development of esophageal cancer (EC). This figure illustrates the intricate interactions between pathogenic bacteria, microbial metabolites, and the host’s immune system, which together drive chronic inflammation, immune dysregulation, and the progression of esophageal cancer. *Porphyromonas gingivalis* and *Fusobacterium nucleatum* are central to this process, activating critical signaling pathways including NF-κB, ERK1/2–Ets1, JAK1/AKT/STAT3, and the NOD1/RIPK2/NLRP3 inflammasome. These pathways lead to the production of pro-inflammatory cytokines (IL-1β, IL-6, IL-8, TNF-α), promoting cell proliferation, the inhibition of apoptosis, and inducing angiogenesis, thereby fostering a tumor-promoting microenvironment. The figure also highlights how microbial dysbiosis which is marked by an increase in harmful bacteria such as *Prevotella*, *Neisseria*, *Eikenella*, and *Aggregatibacter actinomycetemcomitans*, coupled with a decrease in beneficial SCFA-producing bacteria. Dysbiosis exacerbates inflammation and compromises the epithelial barrier. It is further exacerbated by factors such as a low-fiber diet, alcohol consumption, and tobacco use, which together create a milieu conducive to cancer development. Moreover, the direct interaction of these bacteria with the esophageal epithelium, through mechanisms such as TLR-4 activation and E-cadherin binding, promotes epithelial–mesenchymal transition, enhances cancer cell survival, and contributes to chemoresistance. The figure encapsulates the multifaceted role of microbiota in driving EC, from microbial-induced inflammation and immune modulation to direct epithelial transformation, presenting potential therapeutic targets for mitigating EC risk.

**Figure 3 cancers-16-03305-f003:**
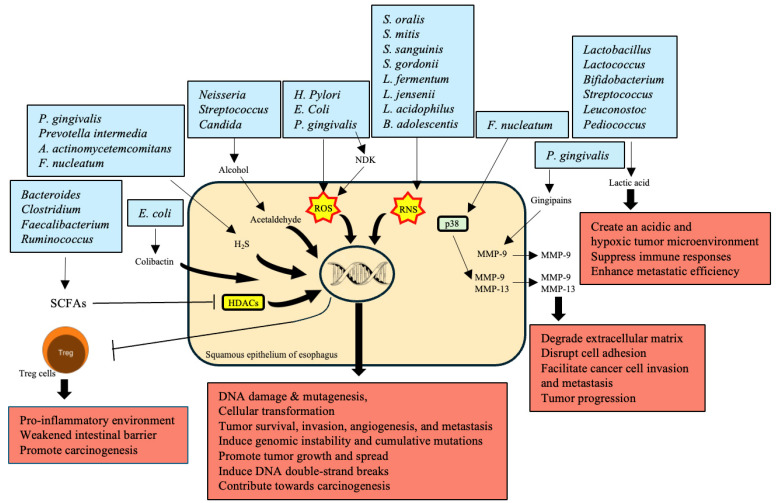
Production of carcinogenic metabolites by microbiota and their role in esophageal cancer (EC) development. The figure illustrates how different types of microbial metabolites trigger the development and progression of esophageal cancer. Anaerobic bacteria such as *Bacteroides*, *Clostridium*, *Faecalibacterium*, and *Ruminococcus* produce short-chained fatty acids. SCFAs are typically helpful and functional in immune and intestinal systems. However, its reduction results in a pro-inflammatory environment, leaky intestine, and promotion of carcinogenesis. *Escherichia coli* generates colibactin, which causes DNA double-strand breaks, thereby affecting genomic instability and cancer. Other types of microorganisms, such as *Neisseria*, *Streptococcus*, and *Candida*, transform alcohol into acetaldehyde, which is a carcinogenic compound and triggers DNA damage and mutagenesis. *Porphyromonas gingivalis*, *Helicobacter pylori*, and *E. coli* produce reactive oxygen species (ROS), while *Streptococcus oralis* and *S. mitis* generate reactive nitrogen species (RNS). Both ROS and RNS irreversibly damage DNA and contribute to chronic inflammation as well as cellular transformation. Moreover, *P. gingivalis* and *F. nucleatum* produce matrix metalloproteinases (MMP-9 and MMP-13), which degrade extracellular matrix and facilitate the invasion of cancer cells and metastasis. In addition, *P. gingivalis* produces H_2_S, which leads to genomic instability and provides a proper environment for tumor growth. Finally, lactic acid is produced by *Lactobacillus*, *Streptococcus*, and related genera. It complicates metabolism by creating an acidic, hypoxic tumor microenvironment, suppressing the immune system and thus supporting metastasis. Therefore, multiple microbial metabolites underlie the development of EC thus serving as potential candidates for pharmacological targeting.

**Figure 4 cancers-16-03305-f004:**
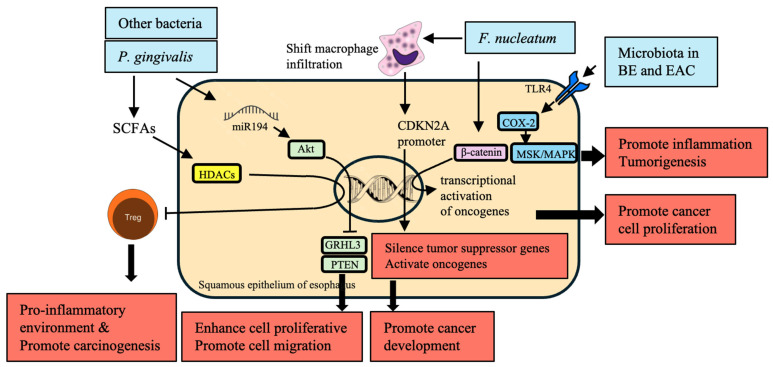
Epigenetic modifications induced by microbiota in the development of esophageal cancer (EC). This figure illustrates how microbiota-induced epigenetic modifications contribute to the development and progression of EC. Short-chain fatty acids (SCFAs) produced by bacteria such as *Porphyromonas gingivalis* and others influence histone deacetylases (HDACs), which, in turn, affect regulatory T (Treg) cell function, promoting a pro-inflammatory environment that supports carcinogenesis. *P. gingivalis* infection also upregulates miR-194 and Akt, while downregulating tumor suppressor genes like grainy head-like transcription factor 3 (GRHL3) and phosphatase and tensin homolog (PTEN), thus enhancing cell proliferation and migration. Additionally, *Fusobacterium nucleatum* influences the methylation status of the cyclin-dependent kinase inhibitor 2A (CDKN2A) promoter, leading to the silencing of tumor suppressor genes and the activation of oncogenes. The activation of Toll-like receptor-4 (TLR-4) in the context of Barrett’s Esophagus and esophageal adenocarcinoma further drives inflammation and tumorigenesis through cyclooxygenase-2 (COX-2) expression and β-catenin signaling. These interactions collectively promote cancer development by altering gene expression profiles, leading to enhanced cancer cell proliferation and migration. This figure highlights the critical role of microbial-induced epigenetic changes in shaping the tumor microenvironment and facilitating esophageal carcinogenesis.

**Figure 5 cancers-16-03305-f005:**
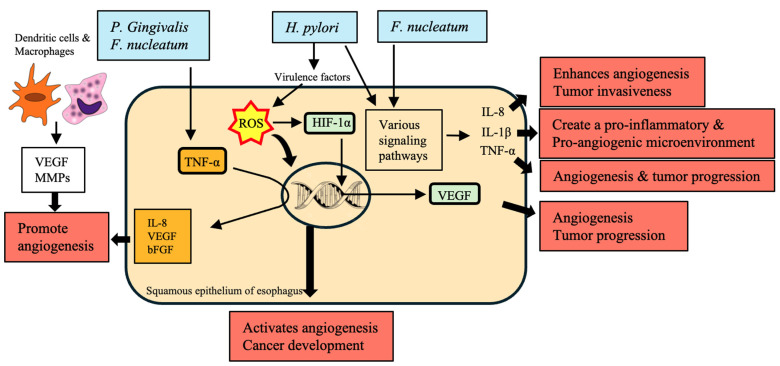
Microbiota-driven angiogenesis in the development of esophageal cancer (EC). This figure illustrates the role of specific bacteria in promoting angiogenesis, a key process mediating the development and growth of EC. This process is mediated by the actions of *Porphyromonas gingivalis*, *Fusobacterium nucleatum*, and *Helicobacter pylori*, which include ROS production as well as stabilization of hypoxia-inducible factor 1-alpha (HIF-1α) that subsequently induces pro-angiogenic factors such as VEGF. The signaling pathways specified by these bacteria, in turn, result in the release of pro-inflammatory cytokines (IL-8, IL-1β, and TNF-α) that promote an even more tumor conducive environment. In the tumor microenvironment (TME), macrophages and dendritic cells promote new blood vessel formation through VEGF and the release of matrix metalloproteinases (MMPs) that break down the extracellular matrix. This angiogenic process is critical for tumor invasiveness, progression, and metastasis. *P. gingivalis* and *F. nucleatum* are highlighted to have the characteristic of enhancing the exacerbation response and accelerating angiogenesis by elevating TNF-α levels along with other angiogenic factors. Additionally, it illustrates the role of H_2_S released from oral bacteria in establishing a pro-angiogenic microenvironment that results in EC progression and dissemination. This figure highlights the intricate interplay between microbial-induced inflammation and angiogenesis that are instrumental to EC management as potential therapeutic targets.

**Table 1 cancers-16-03305-t001:** Comparison of microbial populations in ESCC and control Samples.

Sample	Microbes Increased in ESCC	Microbes Decreased in ESCC or Increased in Control Samples	References
67 paired samples (ESCC tissue vs. non-tumor tissue)	*Fusobacteria* phylum *Fusobacterium* genus	*Firmicutes* phylum *Streptococcus* genus	[41]
32 ESCC samples vs. 21 healthy controls	*Streptococcus* genus *Actinobacillus* genus *Peptostreptococcus* genus *Fusobacterium* genus *Prevotella* genus	*Fusobacteria* phylum *Faecalibacterium* genus *Bacteroides* genus *Curvibacter* genus *Blautia* genus	[42]
32 ESCC samples vs. 15 esophagitis samples	*Streptococcus* genus	*Bacteroidetes* genus *Faecalibacterium* genus *Bacteroides* genus *Blautia* genus	[42]
17 ESCC samples vs. 16 healthy control samples	*Fusobacteria* phylum *Prevotella* genus *Pseudomonas* genus	*Actinobacteria* phylum Ralstonia genus *Burkholderia-Caballeronia-Paraburkholderia* genus	[43]
17 ESCC samples vs. 15 post-op ESCC samples	*Fusobacteria* phylum *Bacteroidetes* phylum *Prevotella* genus	*Pseudomonas* genus	[43]
100 ESCC samples vs. 100 adjacent tissue samples or 30 normal esophagus samples	*P. gingivalis*		[44]
18 ESCC samples vs. 11 normal esophagus samples	*Fusobacteria* phylum *Bacteroidetes* phylum *Spirochaetes* phylum *T. amylovorum*, *S. infantis*, *P. nigrescens*, *P. endodontalis*, *V. dispar*, *A. segnis*, *P. melaninogenica*, *P. intermedia P. tannerae*, *P. nanceiensis*, *S. anginosus*	*Proteobacteria* phylum *Thermi* Phylum	[45]
120 ESCC samples vs. adjacent tissue sample from same subjects	*R. mucilaginosa*, *P. endodontalis* unclassified species in the genus *Leptotrichia* unclassified species in the genus *Phyllobacterium* unclassified species in the genus *Sphingomonas*	class Bacilli *N. subflava* *H. pylori* *A. parahaemolyticus* *A. rhizosphaerae*, unclassified species in the genus *Campylobacter* unclassified species in the genus *Haemophilus*	[46]
60 ESCC samples vs. paired adjacent normal tissue samples	*F. nucleatum*		[47]
54 ESCC samples vs. 4 normal esophageal tissues	*Proteus* genus *Firmicutes* genus *Bacteroides* genus *Fusobacterium* genus		[48]
7 ESCC samples vs. 70 normal control samples (together with 70 esophagitis, 70 low-grade intraepithelial neoplasia and 19 high-grade intraepithelial neoplasia)	*Streptococcus* genus *Haemophilus* genus *Neisseria* genus *Porphyromonas* genus		[49]
48 ESCC samples vs. matched control samples	*Staphylococcus* genus		[50]
111 ESCC samples vs. 41 normal samples	*Bacteroidetes* phylum *Fusobacteria* phylum *Spirochaetae* phylum *Streptococcus* genus *F. nucleatum*	*Butyrivibrio* genus *Lactobacillus* genus	[51]
31 ESCC samples vs. matched controls	*Peptostreptococcaceae*, *Leptotrichia*, *Peptostreptococcus*, *Anaerovoracaceae*, *Filifactor*, *Anaerovoracaceae-Eubacterium_ brachygroup*, *Lachnoanaerobaculum*, *Dethiosulfatibacteraceae*, *Solobacterium*, *Johnsonella*, *Prevotellaceae* UCG_001, and *Tannerella* (higher in N0 stage) *Treponema* and *Brevibacillus* (higher in N1 and N2 stages) *Acinetobacter* (higher in T3 stage) *Corynebacterium*, *Aggregatibacter*, *Saccharimonadaceae*-TM7x, and *Cupriavidus* (higher in T4 stage)		[52]

**Table 2 cancers-16-03305-t002:** Microbiota and mechanisms contributing to chronic inflammation in EC.

Bacteria	Mechanism	Impact on EC	References
*P. gingivalis*	Activates ERK1/2–Ets1 and PAR2/NF-κB pathways	Increased secretion of pro-inflammatory cytokines and chemokines reprogramming TME	[53,54,55]
	Interacts with T cells and macrophages	Disrupts epithelial barrier, induces DNA damage, triggers pro-oncogenic signals	[56]
	LPS activates TLR-4 leading to NF-κB activation	Promotes cell proliferation, inhibits apoptosis, induces angiogenesis through VEGF expression	[57]
	Inhibits HDACs through SCFAs modulating Treg cell function	Supports tumor growth, metastasis, and resistance to therapy	[58,59,60]
*F. nucleatum*	Activates NOD1/RIPK2/NF-κB and NLRP3 inflammasome pathways	Induces high levels of IL-6 and IL-8, driving inflammation-related carcinogenesis	[53,61]
	LPS activates TLR-4 leading to NF-κB activation	Recruits and reprograms immune cells within TME, supporting tumor progression and immune evasion	[62]
	Interacts with T cells and macrophages	Promotes cell proliferation, inhibits apoptosis, induces angiogenesis through VEGF expression	[57]
*E. coli*	Upregulates TLRs 1–3, 6, 7, and 9	Induces early carcinogenic molecular changes through TLR signaling pathway activation	[63]
*A. actinomycetemcomitans*	Produces virulence factors such as leukotoxin and cytotoxic distension toxin	Exacerbates inflammation and cancer risk	[64]

**Table 3 cancers-16-03305-t003:** Microbial dysbiosis: Mechanisms and impact in EC.

Bacteria	Mechanism	Impact on EC	References
*P. gingivalis*	- Activates NF-κB, ERK1/2–Ets1, and PAR2/NF-κB pathways	- Increased production of pro-inflammatory cytokines (IL-1β, IL-6), disruption of epithelial barriers, DNA damage	[53,54,55]
	- Elicits chronic inflammation and immune evasion	- Promotes tumor growth and progression, poor clinical outcomes, potential biomarker for ESCC	[44]
*F. nucleatum*	- Activates NF-κB, NOD1/RIPK2/NF-κB, and NLRP3 inflammasome pathways	- Induction of pro-inflammatory cytokines (IL-6, IL-8), creating a pro-tumorigenic environment	[53,61]
	- Chemokine activation, specifically CCL20	- Aggressive tumor behavior, shorter survival, immune suppression, aiding in tumor progression and metastasis	[47]
	- Utilizes FadA adhesin/invasin to bind E-cadherin, activating β-catenin signaling	- activation of pro-inflammatory cytokines, oncogenes, and stimulation of cancer cell proliferation	[70]
*T. denticola*, *S. anginosus*	- Found in higher abundance in cancerous esophageal tissues	- Production of inflammatory mediators, promotion of an immunosuppressive microenvironment	[71]
*E. coli*	- Upregulates TLRs 1–3, 6, 7, and 9	- Induces early carcinogenic molecular changes through TLR signaling pathway activation	[63]
*Prevotella*	- Produces LPS, activates TLR-4, leading to NF-κB activation	- Promotes chronic inflammation, mucosal barrier disruption, and enhancement of inflammatory milieu	[74]
*Neisseria*	- Produces LPS, activates TLR-4, leading to NF-κB activation	- Promotes chronic inflammation, mucosal barrier disruption, and enhancement of inflammatory milieu	[75,76]
*Eikenella*	- Associated with low fiber intake, leading to increased gram-negative bacteria	- Produces endotoxins that trigger inflammation and promote carcinogenesis	[77]
*A. segnis*, *T. amylovorum*, *P. endodontalis*, *S. infantis*, *V. dispar*, *S. anginosus*, *P. intermedia*, *P. melaninogenica*	- Identified in high-throughput profiling of ESCC	- Contributes to chronic inflammation and tumor-promoting microenvironment	[45]
*Campylobacter*	- Enriched in GERD and BE, associated with IL-18 expression	- Associated with increased expression of carcinogenesis-related cytokines	[78,79]
*Parvimonas*	- Associated with low fiber intake, leading to increased gram-negative bacteria	- Produces endotoxins that trigger inflammation and promote carcinogenesis	[77]
*Leptotrichia*	- Observed in GERD and BE patients	- Produces pro-inflammatory molecules, exacerbating mucosal damage and inflammation, contributing to progression to EAC	[80]
*Lautropia*, *Bulleidia*, *Catonella*, *Corynebacterium*, *Moryella*, *Peptococcus*, *Cardiobacterium*	- Lower carriage in ESCC patients compared to controls	- Altered saliva microbiota associated with higher risk of ESCC	[81]
*Tannerella forsythia*	- Increased levels in EC patients	- Associated with higher risk of EAC	[7]

**Table 4 cancers-16-03305-t004:** Microbiota and production of carcinogenic metabolites in EC.

Bacteria	Mechanism	Impact on EC	References
*Bacteroides*, *Clostridium*, *Faecalibacterium*, *Ruminococcus*	Produce SCFAs like butyrate, acetate, and propionate through dietary fiber fermentation	Reduced SCFA production contributes to a pro-inflammatory environment and weakened intestinal barrier, promoting carcinogenesis	[92]
*Neisseria*, *Streptococcus*, *Candida*	Metabolize alcohol into acetaldehyde, a highly toxic and carcinogenic substance	Causes DNA damage, mutagenesis, and gut microbiota disruption, increasing EC risk	[95]
*P. gingivalis*, *H. pylori*, *E. coli*	Produce ROS	Leads to DNA damage, cellular transformation, tumor survival, invasion, angiogenesis, and metastasis	[96,97]
*S. oralis*, *S. mitis*, *S. sanguinis*, *S. gordonii*, *L. fermentum*, *L. jensenii*, *L. acidophilus*, *B. adolescentis*	Produce RNS	Contribute to DNA damage and cancer progression through nitrosative stress	[98,99]
*P. gingivalis*, *F. nucleatum*	Overexpress MMPs; *P. gingivalis* produces gingipains to activate MMP-9; *F. nucleatum* stimulates MMP-9 and MMP-13 through p38 signaling	Degrade extracellular matrix, disrupt cell adhesion, facilitating cancer cell invasion and metastasis, critical in tumor progression	[55,100]
*P. gingivalis*, *Prevotella intermedia*, *A. actinomycetemcomitans*, *F. nucleatum*	Produce H_2_S, a genotoxic volatile sulfur compound	Induces genomic instability and cumulative mutations, promoting tumor growth and spread by activating various signaling pathways	[101,102]
*Lactobacillus*, *Lactococcus*, *Bifidobacterium*, *Streptococcus*, *Leuconostoc*, *Pediococcus*	Produce lactic acid through fermentation	Overproduction creates an acidic and hypoxic tumor microenvironment, suppressing immune responses and enhancing metastatic efficiency	[103]
*E. coli*	Secretes colibactin, a metabolic genetic toxic substance	Induces DNA double-strand breaks, leading to genomic instability and contributing significantly to carcinogenesis	[104]

**Table 5 cancers-16-03305-t005:** Microbiota and their direct interaction mechanisms with epithelial cells in EC.

Bacteria	Mechanism	Impact on EC	References
*P. gingivalis*	Activates ERK1/2–Ets1 and PAR2/NF-κB pathways	Promotes proliferation, migration, and invasion of epithelial cells	[53,54]
	Induces antiapoptotic activity via JAK1/AKT/STAT3 pathway	Reduces apoptotic activity of epithelial cells	[114]
	Secretes NDK	Enhances BCL2 to BAX ratio	[106]
	Accelerates S-phase progression by manipulating CDK activity	Promotes cancer cell proliferation	[115]
*F. nucleatum*	Activates NOD1/RIPK2/NF-κB pathway	Enhances ESCC cell growth and migration	[53,61]
	Influences TME through chemokine activation	Associated with shorter survival times and aggressive tumor behavior	[116,117]
	Activates TLR-4	Promotes β-catenin signaling leading to oncogene activation	[70,118]
	Binds to E-cadherin on carcinoma cells	Facilitates cancer cell proliferation	[70]
*Campylobacter*, *Leptotrichia*, *Rothia*, *Capnocytophaga*	Enriched in GERD and BE	Contributes to chronic inflammation and epithelial cell transformation	[79,119]
*A. actinomycetemcomitans*	Produces virulence factors that interact with epithelial cells	Promotes cell transformation and carcinogenesis	[64]
*T. denticola*, *S. mitis*, *S. anginosus*	Dominates microbiota in cancerous esophageal tissues	Suggests direct interaction with epithelial cells contributing to disease progression	[71]
*Candida*, *Neisseria*	Metabolizes alcohol into acetaldehyde	Causes DNA damage, mutagenesis, and disrupts gut microbiota	[120,121]

**Table 6 cancers-16-03305-t006:** Microbiota and epigenetic modifications in EC.

Bacteria	Mechanism	Impact on EC	References
*P. gingivalis*	- Inhibits HDACs through SCFAs, modifying Treg cell function and numbers	- Creates a pro-inflammatory environment, contributing to carcinogenesis	[58,60]
	- Upregulates miR-194 and Akt, downregulates GRHL3 and PTEN	- Enhances pro-proliferative and pro-migratory phenotype of esophageal tumors	[132]
*F. nucleatum*	- Alters macrophage infiltration and methylation of the CDKN2A promoter	- Silences tumor suppressor genes and activates oncogenes, promoting cancer development	[133]
	- Activates β-catenin signaling, leading to transcriptional activation of oncogenes	- Promotes cancer cell proliferation through activation of oncogenic pathways	[70,134]
Microbiota in General	- Produces SCFAs that inhibit HDACs, impacting immune response and inflammation	- Creates a pro-inflammatory environment, contributing to carcinogenesis	[60]
	- Interacts with epithelial cells, leading to genetic changes in mRNAs, miRNAs, and LncRNAs	- Disrupts normal cell regulatory mechanisms, promoting malignant transformation	[135,136]
Microbiota in BE and EAC	- Activates TLR-4, influencing COX-2 expression through NF-κB-independent pathways like MSK and MAPK	- Leads to modifications in gene expression that promote inflammation and tumorigenesis	[123,131]

**Table 7 cancers-16-03305-t007:** Microbiota and interaction with GERD: Mechanisms and impact in EC.

Bacteria	Mechanism	Impact on EC	References
*Veillonella*, *Prevotella*, *Neisseria*	Produces LPS, activates TLR-4 leading to NF-κB activation	Creates a pro-inflammatory environment, contributing to carcinogenesis	[142,143]
*Streptococcus*	Increases prevalence with age, producing pro-inflammatory cytokines	Influences chronic inflammation and increases the risk of EC	[119]
*H. pylori*	- Causes chronic gastritis, leading to changes in gastric acid secretion and subsequent GERD	- Promotes the progression of GERD to BE and EAC	[144]
*Campylobacter*	- Enrichment in GERD and BE patients, associated with inflammatory responses	- Contributes to chronic inflammation and changes in the esophageal mucosa, promoting the progression to EAC	[145]
*F. nucleatum*	- Adheres to and invades epithelial cells, modulates immune response, and promotes inflammation	- Exacerbates progression of BE to EAC through TLR activation and promoting an oncogenic microenvironment	[146]
*Prevotella*	- Increased prevalence in the esophageal microbiota of GERD patients, known for its role in inflammatory processes	- Leads to chronic inflammation and mucosal damage, fostering conditions conducive to BE and EAC	[147]
*S. anginosus*	- Associated with the esophageal microbiota in GERD and BE, contributing to chronic inflammation	- Promotes epithelial cell alterations, facilitating progression from GERD to BE and EAC	[71,148]
*Leptotrichia*	- Enrichment in GERD and BE patients, associated with inflammatory responses	- Promotes chronic inflammation and epithelial cell transformation, contributing to carcinogenesis	[149,150]
*Rothia*	- Enrichment in GERD and BE patients, associated with inflammatory responses	- Contributes to chronic inflammation and mucosal damage, facilitating the progression to EAC	[151]
*Capnocytophaga*	- Enrichment in GERD and BE patients, associated with inflammatory responses	- Promotes chronic inflammation and changes in the esophageal mucosa, fostering conditions conducive to EAC	[152]

**Table 8 cancers-16-03305-t008:** Microbiota and metabolic changes in EC.

Bacteria	Mechanism	Impact on EC	References
*Bacteroides*, *Clostridium*, *Faecalibacterium*, *Ruminococcus*	Produce SCFAs, modulate inflammation	Maintain gut health; reduced SCFA production leads to a pro-inflammatory environment and cancer risk	[89,92]
*H. pylori*	Induces chronic gastritis, alters gastric acid secretion	Promotes GERD, BE, and EAC	[164]
*Campylobacter*	Induces inflammatory responses	Promotes chronic inflammation and progression to BE and EAC	[46]
*Lactobacillus*, *Streptococcus*, *Bifidobacterium*, *Leuconostoc*	Produce lactic acid, create low pH hypoxic environment, induce Warburg effect	Immunosuppression, enhanced tumor metastasis, support cancer cell survival and proliferation	[80]
*F. nucleatum*	Produces LPS, activates β-catenin signaling, enhances oncogene expression (C-myc, cyclin D1)	Promotes cancer cell proliferation, chronic inflammation, and carcinogenesis	[134]
*P. gingivalis*	Modulates ATP/P2X7 signaling, affects ROS and antioxidant responses	Contributes to cancer development through ROS-mediated DNA damage and inflammatory responses	[106]
*Streptococci*, *Candida* yeasts	Metabolize alcohol to acetaldehyde via ADH activity	Causes DNA damage, increases carcinogenesis risk	[95]

**Table 9 cancers-16-03305-t009:** Microbiota and mechanisms of angiogenesis in EC.

Bacteria	Mechanism	Impact on EC	References
*H. pylori*	Increases ROS production through virulence factors	Activates angiogenesis and cancer development	[168]
	Promotes hypoxic conditions stabilizing HIF-1α	Upregulates pro-angiogenic genes such as VEGF, contributing to tumor progression and poor prognosis	[169]
*F. nucleatum*	Influences IL-8 production	Enhances angiogenesis and tumor invasiveness	[113]
	Enhances IL-1β production	Creates a pro-inflammatory and pro-angiogenic microenvironment	[170]
	Increases TNF-α levels	Contributes to angiogenesis and tumor progression	[66]
	Activates β-catenin signaling, enhancing β-catenin, C-myc, and cyclin D1 expression	Enhances cancer cell proliferation and tumor growth	[70]
*P. gingivalis*	Modulates inflammatory responses and cytokine production	Enhances tumor angiogenesis	[170]
	Increases TNF-α levels	Promotes cancer cell proliferation and metastasis	[171]
	Produces H_2_S, activating proliferation, migration, and invasive signaling pathways	Contributes to a hypoxic, pro-angiogenic microenvironment	[101]
*Streptococcus* species	Stimulates the production of angiogenic factors such as IL-8, VEGF, and bFGF	Promotes angiogenesis and cancer cell growth	[172]
General oral microbiota	Produces IL-1β, which activates endothelial cells to produce VEGF and other pro-angiogenic factors	Provides an inflammatory microenvironment conducive to angiogenesis and tumor progression	[173,174]

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
