# Peer review of "Mechanistic Insights on Microbiota-Mediated Development and Progression of Esophageal Cancer"

_cancers, 2024, doi:10.3390/cancers16193305_

Round 1

Reviewer 1 Report

Comments and Suggestions for Authors

The manuscript comprehensively described the factors of esophageal cancer formation, explained the intricate connection between microbiome and esophageal tumorigenesis and particularly pointing out the cell signaling pathways, which is essential and important for overviewing the whole research field. Here are my concerns that could be helpful to improve this review.

I would suggest to combine and shorten some parts which have the similar and overlapping mechanism, such as the chronic inflammation and interaction with GERD. In general, chronic GERD causes the inflammation in the upper-gastrointestine and the mechanisms are quite similar. Also, the angiogenesis part demonstrated the same mechanism (ROS and TNFa signaling) compared with chronic inflammation and production of carcinogen. Therefore, authors should shorten the overlapping content for a concise review.

It is reasonable and understandable for the interaction between esophageal epithelial cells and microbiota via the production of metabolites. How do microbiota directly connect with the epithelial cells and consequentially activate cell signaling pathway? Authors only summarized the downstream pathways but not clearly described the processing of the direct interaction.

Comments on the Quality of English Language

English language issue is minor.

Reviewer 2 Report

Comments and Suggestions for Authors

In this manuscript, the authors have provided a collective information on the role of microbiota in esophageal cancer. It's an interesting topic with clinical significance. The authors should include the following points to improve the manuscript.

1. A separate section on the pathogenesis of GERD and BE will help the readers to understand the significance of microbiota in early stages of EAC.

2. A separate figure on the mechanism of reflux-induced changes in microbiota and its tumorigenic outcomes would improve the manuscript.

Comments on the Quality of English Language

Moderate English editing can improve the readability.

Round 2

Reviewer 2 Report

Comments and Suggestions for Authors

The manuscript was improved by revision and can be accepted for publication.

Comments on the Quality of English Language

Minor English language editing will improve the readability.